# Topological Distribution of the Sex Hormone Receptor Expressions Highlights the Importance of Stromal ERα and Epithelial PR in Malignant Transformation of the Uterine Cervix

**DOI:** 10.3390/ijms26094418

**Published:** 2025-05-06

**Authors:** Mun-Kun Hong, Jen-Hung Wang, Ming-Hsun Li, Cheng-Chuan Su, Chiu-Hsuan Cheng, Tang-Yuan Chu

**Affiliations:** 1Department of Obstetrics and Gynecology, Hualien Tzu Chi Hospital, Hualien 97004, Taiwan; jeff06038@gmail.com; 2Institute of Medical Sciences, Tzu Chi University, Hualien 97004, Taiwan; 3Center for Prevention and Therapy of Gynecological Cancer, Hualien Tzu Chi Hospital, Hualien 97004, Taiwan; 4School of Medicine, Tzu Chi University, Hualien 97004, Taiwan; 5Department of Medical Research, Hualien Tzu Chi Hospital, Hualien 97004, Taiwan; 6Department of Pathology, Hualien Tzu Chi Hospital, Hualien 97004, Taiwan; 7Departments of Clinical Pathology and Anatomic Pathology, Dalin Tzu Chi Hospital, Dalin, Chiayi 62247, Taiwan

**Keywords:** cervical cancer, stroma, malignant transformation, metastasis, progesterone receptor B

## Abstract

To investigate the changes of ERα and PRs in the epithelium and stroma of normal and neoplastic uterine cervix. Two pathologists independently scored the expression levels of ERα, PR(A+B), and PRB in the stroma and epithelium of normal, cervical intraepithelial neoplasia grade 2 and 3 (CIN2/3), carcinoma in situ (CIS), and invasive cervical carcinoma (ICC) specimens. Sex hormone receptors were abundantly expressed in the stroma compared to the epithelium or carcinoma of the cervix. Stromal ERα was progressively upregulated during cervical carcinogenesis, with an immunoreactive score (IRS) of 1.3 ± 1.5, 2.1 ± 1.9, and 3.6 ± 3.3 in the CIN2/3, CIS, and ICC groups, respectively (*p* < 0.001). By contrast, epithelial PR(A+B) and PRB were downregulated, with IRS of 0.4 ± 0.7 and 0.5 ± 0.8, 0.1 ± 0.4 and 0.2 ± 0.6, and 0.1 ± 0.6 and 0.1 ± 0.4 in the CIN2/3, CIS, and ICC groups, respectively (*p* < 0.001). During the CIN2/3 transition, the coexpression relationship between ERα and PRs began to break down. Although epithelial PR(A+B) was downregulated, stromal PR(A+B) and PRB were upregulated with IRS of 2.0 ± 2.0 and 2.0 ± 1.9 as well as 2.1 ± 2.3 and 3.2 ± 3.2 in the CIS (*p* = 0.009) and ICC groups (*p* < 0.001), respectively. After complete transformation, the stromal PRB was significantly upregulated, and its loss was related to more distant metastasis and poorer prognosis. The results of this study highlight the carcinogenic role of stromal ERα, the tumor suppressor role of epithelial PRs, and the importance of stromal PRB in the development of cervical cancer; they can be used as a basis for developing prevention and treatment strategies for this disease.

## 1. Introduction

Cervical cancer development is strongly linked to reproductive factors. Research on age-specific incidence rates in populations without screening programs indicates that cervical cancer incidence begins to rise after menarche, increases steadily until menopause, peaks shortly thereafter, and then gradually declines [1]. Postmenopausal new development of cervical cancer is rare [1]. A large-scale epidemiological study identified a high number of full-term pregnancies and prolonged hormonal contraceptive use as independent risk factors, highlighting the significant role of sex hormones in cervical carcinogenesis [2].

Most studies examining estrogen receptor α (ERα) and progesterone receptor (PR) expression in cervical cancer were conducted in the 1970s and involved a limited number of cases. These studies consistently reported ERα and PR expression in the less differentiated layers of the squamous epithelium, with minimal fluctuations during the menstrual cycle [3,4,5]. An analysis of neoplastic epithelium found no significant association between receptor expression and neoplasm severity, leading to inconclusive findings regarding their prognostic relevance in cervical cancer [6]. Research on ERα and PR expression in the stroma of normal and neoplastic epithelium remains limited [7]. One study reported variable ERα and PR expression in stromal cells of the uterine cervix, independent of the menstrual cycle. Our previous study was the first to demonstrate that ERα and PRB expression in tumor stroma predicts a favorable prognosis in a cohort of 95 cervical cancer patients followed for over five years [8,9].

Studies using mouse models have provided valuable insights into the role of ERα and PR in cervical carcinogenesis. Chung et al. [10] demonstrated that ERα and chronic estrogen exposure at physiological levels are essential for tumorigenesis in K14-HPVE6 or -HPVE7 transgenic mice, which developed cervical cancer with nearly 100% efficiency by 12 months, while the removal of exogenous estrogen prevented the development of cervical neoplasia and invasive cervical carcinoma (ICC). In contrast, estrogen antagonist use led to regression of the pre-existing cervical neoplasia [11]. PR functions as a ligand-dependent tumor suppressor in cervical cancer. In transgenic mice with cervical intraepithelial neoplasia (CIN) lesions, under the induction of PR by estrogen [12], treatment with medroxyprogesterone acetate (MPA) was shown to prevent cervical cancer development. This finding suggests that progesterone and PR may act as tumor suppressors and offer therapeutic potential for CIN [13].

To further understand the role of female sex hormones in cervical cancer development, this study examined sex hormone receptor expression in normal and neoplastic uterine cervix tissue across different severities of cervical neoplasia. Specifically, it aimed to elucidate the topological distribution of these receptors in the epithelium/carcinoma and stroma of the uterine cervix, ranging from normal tissue to CIN grades 2 and 3 (CIN2/3), carcinoma in situ (CIS), and invasive cervical carcinoma (ICC).

## 2. Results

### 2.1. Predominant Expression of Sex Hormone Receptors in the Stroma of the Cervix with Progressive ERα Increase During Malignant Transformation

We performed a comparative analysis of the expression levels of ERα, PR(A+B), and PRB in the epithelial and stromal components of the cervix using 58 normal, 44 CIN2/3, 70 CIS, and 159 ICC specimens (Figure 1, Table 1). Across all four groups, the three receptors were predominantly expressed in the stroma (Table 1). The IRS and positive rate of ERα in the stroma and epithelium of normal, CIN2/3, CIS, and ICC specimens were 0.9 ± 1.1 (51.7%) vs. 0.5 ± 0.7 (30.9%), 1.3 ± 1.5 (65.1%) vs. 0.8 ± 1.2 (35.3%), 2.1 ± 1.9 (81.4%) vs. 1.1 ± 1.4 (61.4%), and 3.6 ± 3.3 (82.8%) vs. 0.8 ± 1.7 (28.6%), respectively (Table 1). As shown in Figure 2, a progressive increase in ERα expression was observed in the stroma throughout the process of cervical malignant transformation.

By contrast, the ERα expression in the epithelium of the normal cervix was only observed in a small portion of cases at low levels. However, it slightly increased in CIN2/3 and CIS specimens. Notably, during the CIS to ICC transition, the positive rate significantly decreased from 61.4% to 28.6% (Table 1).

### 2.2. Downregulation of PRs in the Epithelium During CIS to ICC Transition

The positivity rate of PR(A+B) and PRB in the cervical epithelium exhibited a significant decrease during CIN2/3–CIS transition (from 29.7% to 8.8% and 29.7% to 15.7%, respectively) and further decreased to 4.1% and 7.7% in ICC, respectively (*p* < 0.001) (Table 1, Figure 2). During the process of cervical malignant transformation, the mean IRS decreased from 0.4 ± 0.7, 0.1 ± 0.4, to 0.1 ± 0.6 for PR(A+B) and from 0.5 ± 0.8, 0.2 ± 0.6, to 0.1 ± 0.4 for PRB, respectively.

### 2.3. Age-Related Decrease in Stromal PRs Across Disease States, Except in CIS

We determined whether the expression of hormone receptors changes with age (Figure 3A–C). In the stroma, there was a statistically non-significant trend in age-related reduction in PRB levels in normal and CIN2/3 specimens. This trend was lost in CIS but became statistically significant and reappeared in ICC (Figure 3C). The age-related changes in PRs were pronounced, particularly for PRB in ICC. Pearson’s correlation coefficient (r) decreased from 0.226 and 0.296 in the normal and CIN2/3 specimens to 0.087 in CIS specimens and then increased to 0.215 in ICC specimens (*p* = 0.008).

Considering the effect of age, we stratified the cohort into premenopausal (<50 years old) and menopausal (≥50 years old) groups. Figure 4 and Appendix A shows that stromal PR(A+B) and PRB expression was significantly lower after menopausal age in each disease category. However, the difference was less pronounced for ERα.

### 2.4. Disruption of ERα–PR Expressional Correlation During CIN2/3 of Cervical Carcinogenesis

The PR gene encodes two isoforms of the PR protein: PRA and PRB. The E2-bound ERα upregulates the expression of both isoforms by acting on the PR promoter [14,15]. Indeed, the stromal expression levels of ERα and PRs were highly correlated in individual normal, CIS, and ICC specimens, with correlation coefficients of 0.528, 0.607, and 0.745, respectively (*p* < 0.001 for each). However, in CIN2/3 specimens, the ERα/PR correlation was disrupted (r = 0.292, *p* = 0.072) (Table 2).

In the epithelium, the correlation between ERα and PR was consistently strong in the normal cervix (r = 0.555, *p* < 0.001). However, this association weakened during malignant transformation, with varying PR(A+B) and PRB rates. The ERα/PR(A+B) correlation decreased starting from CIS (r = 0.288, *p* = 0.17) and was completely lost in ICC (r = 0.047, *p* = 0.572). By contrast, the ERα/PRB association gradually diminished from normal tissue, CIN2/3, CIS, to ICC, with r values of 0.555, 0.335, 0.389, and 0.101, respectively.

### 2.5. Disruption of the Topological Epithelium–Stroma Association of Sex Hormone Receptor Expression During Malignant Transformation

The interaction between the epithelial and stromal compartments is crucial for the remodeling and function of the uterine cervix during pregnancy and the menstrual cycle. Therefore, we examined the topological relationship of ER/PR expression in the epithelium and stroma of cervixes across the normal and different cervical neoplasia groups. We found a strong epithelium–stroma link of ERα throughout disease progression. About PRB, the association was robust in normal tissues (r = 0.424, *p* = 0.002), significantly weakened in CIN2/3 (r = 0.282, *p* = 0.087), recovered in CIS (r = 0.377, *p* = 0.004), and completely disrupted in ICC (r = −0.002, *p* = 0.980). In terms of PR(A+B), the topological relationship significantly weakened from normal tissue (r = 0.478, *p* < 0.001) to CIN2/3 (r = 0.046) and remained low in CIS (r = −0.056) and ICC (r = 0.017) (Table 3).

## 3. Discussion

This study provides a comprehensive overview of the temporal and topological changes in ERα/PR expression in the epithelium and stroma of the uterine cervix throughout the malignant transformation process (Figure 5). We found that sex hormone receptors are highly expressed in the stroma, a detail often overlooked in previous research, highlighting their critical role in the malignant transformation process.

The key findings include a progressive upregulation of stromal ERα during cervical carcinogenesis, in contrast with a downregulation of PRs in the transformed epithelium of CIS and ICC. The stromal PRB showed an early increase from normal tissue to CIN2/3, a decrease during CIN2/3 to CIS transition, and a rebound at the ICC stage.

The coexpression relationship between ERα and PR was disrupted during the actively transforming CIN2/3 stage in the epithelium and stroma, and a further disruption was observed in the stroma at the ICC stage. Notably, the topological relationship of hormone receptor expression in the epithelium and stroma remained strong for ERα across all disease stages. In contrast, it was diminished in CIN2/3 and completely lost in ICC for PRB. About to PR(A+B), significant disruptions were observed in CIN2/3 and more advanced stages.

This study revealed for the first time a progressive upregulation of stromal ERα during cervical carcinogenesis, with levels increasing fourfold from normal cervix to ICC (Figure 1). Estrogen exposure has been recognized as an independent risk factor in cervical cancer development. A nationwide, population–based study demonstrated that the use of antiestrogen is associated with a significant reduction in cervical dysplasia [16]. According to transgenic studies involving HPVE6/E7 transformation of the mouse cervix, estradiol and ERα play critical roles throughout the carcinogenic process, from early atypical lesions to dysplasia and invasive carcinomas [8,9]. Notably, ERα signaling in the stroma, rather than in the epithelium, is essential for carcinogenesis in transgenic mice [10,17]. Although ERα levels in the epithelium of CIN2/3 and CIS also exhibited increases, a contrasting decrease was observed during CIS to ICC progression. This decline suggests an invasion-associated downregulation, which is likely influenced by paracrine signaling from the stroma.

In the transition from normal tissue to CIN2/3, there was a significant increase in PR(A+B) and PRB expression levels in the stroma. A previous study found that progesterone is induced in mice infected with various DNA and RNA viruses, where it activates downstream antiviral genes and enhances innate antiviral responses in immune cells [18]. Furthermore, an epidemiological study found that the use of Depo-Provera (MPA, a synthetic form of progesterone) was linked to a reduced risk of cervical cancer among women infected with human papillomavirus (HPV) [19]. In a transgenic HPV-induced cervical cancer mouse model, MPA treatment effectively prevented cancer development, which was characterized by decreased cell proliferation and increased apoptosis in CIN lesions. This protective effect was absent when PR expression was genetically eliminated [13]. Whether progesterone may exert an antiviral effect through its action on PRs in the stroma during the early stages of cervical transformation with persistent HPV infection (Figure 5) is worth further investigation.

The development of the Müllerian epithelium is regulated by estrogen and progesterone through paracrine mechanisms, which necessitate the crosstalk of their receptors across the basement membrane [20,21]. The PR changes in the stroma of the transforming cervix may reflect different needs in multistage carcinogenesis. It is speculated that in the normal to CIN2/3 stage, stromal PR increases to combat HPV infection and transformation. In the CIN2/3–CIS progression stage, a decrease in PR is required to advance the transformation. We observed a PR decrease in the stroma and epithelium during this transition (Figure 2). In the CIS stage, wherein the full thickness of the epithelium is transformed, the new topological interaction between the epithelium and stroma is related to invasion and anti-invasion activities, respectively. Meanwhile, the oncogenic ERα in the stroma further increases (Figure 5). However, further studies are warranted to prove this speculation.

Epithelial PR reportedly plays a tumor suppressor role in a transgenic mouse model of cervical carcinogenesis. The deletion of one or both PGR alleles in the cervical epithelium promoted spontaneous CIN and ICC, with all lesions showing low PR expression [22]. Our IHC study of the human cervix echoes this finding, demonstrating a progressive downregulation of epithelial PR throughout the malignant transformation process (Figure 4 and Figure 5). It is also found that epithelial ERα expression is gradually decreased as the disease progresses to cancer [23]. In the present study, epithelial ERα expression is gradually increased from normal, CIN2/3, to CIS and then decreased at CIS-ICC progression. The difference may be explained by different sample sizes, methodologies, and patients’ menstrual cycle statuses [3]. Regardless, both studies show an ESR downregulated in HSIL-ICC progression, and ERα in ICC and precursor cell lines reportedly plays an anti-invasive role. The knockdown of the ESR1 gene resulted in a more invasive phenotype in cells with high ERα expression. In contrast, the restoration of the ESR1 gene decreased cell invasion in cells with low ERα expression [23].

In this study, we found that the expressions of ERα and PRs in the epithelium and stroma of the cervix are highly correlated. This finding is consistent with the transactivating function of estrogen-bound ERα at the PGR promoter [24]. The coexpression relationship between ERα and PRs was broken in the epithelium and stroma of CIN2/3 and also in the carcinoma part of ICC. Moreover, we discovered that although cervical carcinomas maintained consistent levels of ERα across different ages, stromal PRB expression seemed to exhibit a steady decline with advancing age at diagnosis. This finding suggests a non-genomic, age-related control of PR expression in the ICC epithelium and CIN2/3 epithelium and stroma. These alterations may play biological roles in facilitating malignant transformation and cancer invasion within the cervical epithelium–stroma microenvironment.

A recent study observed a significant association between stromal PRB expression and less distant metastasis in ICC [9]. In an ICC cohort of 169 cases with >10 years of follow-up, we found that stromal PRB independently conferred a lower risk (hazard ratio 0.39, 95% CI: 0.18–0.87, *p* = 0.022) of 5-year mortality considering age, histology, FIGO stage, tumor differentiation, and lymphatic and hematogenous metastasis. In particular, stromal PR expression was associated with a lower rate of hematogenous distant metastasis (*p* = 0.011). We carefully observed an increase in stromal PRB in the CIS to ICC stage of cervical cancer and found an approximately 50% increase on average. When stratified by distant metastasis status, the median IRS was significantly higher in the nonmetastatic group (3.24 ± 3.18) than in the metastatic group (2.40 ± 3.71, *p* = 0.048) [9]. Thus, in the ICC cervix, PRB tends to increase for fighting against cancer metastasis, whereas a downregulation of PRB due to unknown mechanisms facilitates distant metastasis (Figure 6 and Figure 7).

The findings of the present study suggest that cervical carcinogenesis is not purely an epithelial process but involves significant stromal reprogramming and disruption of epithelial–stromal hormonal crosstalk. The alteration in ERα and PR expression patterns reflects a shift from hormone-regulated homeostasis to a dysregulated microenvironment that supports malignant progression. This has the following potential implications: first, therapeutic targeting of stromal ER signaling; second, biomarker development for early detection or progression risk; third, understanding age- and hormone-related risk factors; and lastly, potential roles for hormonal manipulation (e.g., progestins) in prevention or treatment.

This study has several strengths. To the best of our knowledge, this study is the first to comprehensively examine ERα and PR expression levels in the epithelium and stroma of the cervix across various stages of cervical neoplasia. The histological examination was performed meticulously, with paraffin blocks being sectioned by a senior pathologist to ensure accurate diagnosis and evaluation of the stroma. Moreover, ERα and PR expression levels were independently assessed by two senior gynecological pathologists from different hospitals, enhancing the reliability of the results. We also deliberately chose to exclude CIN1 cases, which are typically transient, thereby focusing on more relevant stages of cervical carcinogenesis.

However, there are limitations to consider. One significant limitation is the inability to specifically detect the PRA isoform because of the lack of a dedicated antibody, which restricted the study to measuring the combined PR(A+B) expression levels. Furthermore, the lack of analysis of HPV status may limit the understanding of how HPV may influence the sex hormone receptor expression in cervical neoplasia. Our previous study found that ERα/PR expression in the stroma or epithelium of the ICC cervix was unrelated to HPV infection status [7,8]. A previous study also showed no difference in stroma ERα expression in HPV-positive and HPV-negative normal and CIN cervixes [7,24]. Lastly, this study was conducted in a specific cohort with defined inclusion criteria, so the findings may not be generalizable to all populations or settings.

## 4. Materials and Methods

### 4.1. The Patient Samples

This study comprehensively examined formalin-fixed paraffin-embedded (FFPE) tissue samples, including CIN2/3, CIS, and ICC. The Research Ethics Committee of Hualien Tzu Chi Hospital, Hualien, Taiwan, approved the two protocols of this study. ICC specimens were retrieved from patients with cervical cancer who underwent radical hysterectomy at Hualien Tzu Chi Hospital between 2000 and 2010 (IRB104-151-A). Normal cervical specimens were obtained from patients who underwent total hysterectomy because of uterine myoma or adenomyosis. CIN2/3 and CIS specimens were collected during cervical biopsies and conizations at Hualien Tzu Chi Hospital from 2015 to 2017 (IRB104-100-A). A senior pathologist sectioned these paraffin blocks to confirm the diagnoses and adequacy of the stromal part. Cases that did not meet the diagnoses or lacked an adequate stromal component were excluded from the study. Clinical characteristics, including patient’s age, parity, and menopausal status, were also reviewed in the study.

### 4.2. Immunohistochemistry (IHC)

The expression levels of ERα, PR(A+B), and PRB in FFPE primary cervical cancer tissues were analyzed using IHC. The specimens were cut into 5 μm sections and mounted on slides. After deparaffinization in xylene, the slides were rehydrated through a graded alcohol series and placed in running water. Immunohistochemical detection was performed using the Novolink Polymer Detection System (Leica Biosystems, Novocastra Laboratories, Newcastle upon Tyne, UK). Briefly, antigen retrieval was performed by heating the slides in 10 mM citrate buffer (pH 6.0). Then, the slides were incubated with a peroxidase block to neutralize endogenous peroxidase activity. Subsequently, the slides were treated with either anti-ERα monoclonal antibody (dilution 1:250; ab108398; Abcam, Singapore Pte Ltd., 11 North Buona Vista Drive, #16-08,The Metropolis Tower Two, Singapore, 138589) or anti-PR antibody (PRB-specific YR85; dilution 1:100; ab32085; Abcam; PRA-specific NCL-L-PGR-312, which was found also binds PRB, hence marked as PR(A+B) [25,26]; dilution 1:100; PGR-312-L-F; Leica Biosystems, 21440 W. Lake Cook RoadFloor 5, Deer Park, IL 60010, USA) for 30 min [25,27]. Next, the slides were incubated with the Novolink polymer and then treated with DAB chromogen solution to visualize peroxidase activity. Each IHC assay was performed in duplicate or triplicate, depending on the amount of tissue available.

### 4.3. Staining Evaluation

Two experienced histopathologists from two hospitals who were blinded to the clinical characteristics independently performed a staining evaluation. The staining of ERα, PR(A+B), and PRB in tumor cells and adjacent stroma was performed. Although polymorphonuclear cell infiltration is uncommon in ERα/PR evaluation, this potential interference was addressed by examining areas with minimal polymorphonuclear cell presence. The results were evaluated and scored using the immunoreactive score (IRS) [28]. The IRS is reproducible, definable, and widely recognized as the “gold standard” scoring system. This system has been used for cervical cancer [8,9] and various types of malignant gynecological tumors [14,15] and has been recommended by leading pathology organizations [16,29]. The percentage of positively stained cells was scored as follows: 0 (0%), 1 (1–10%), 2 (11–50%), 3 (51–80%), or 4 (>80%). Meanwhile, the staining intensity was scored as follows: 0 (none), 1 (weak), 2 (moderate), or 3 (strong). Multiplying these two scores yielded an IRS of 0 to 12. In this study, the mean IRS from the two histopathologists for every FFPE cervical tissue was used for analysis.

### 4.4. Statistical Analyses

An independent two-sample *t*-test or one-way analysis of variance was used to detect the mean difference between different groups. Categorical variables were compared using the Chi-squared or Fisher’s exact test. The Cochran–Armitage trend test was used to evaluate the underlying trends. Pearson’s correlation coefficient was used to assess the linear relationship between two covariates. All statistical analyses were performed using SPSS software (version 17.0; SPSS Inc., Chicago, IL, USA).

## 5. Conclusions

This study offers new insights into the roles of sex hormone receptors and their interactions between the epithelium and stroma during the development of cervical carcinomas from precursor stages. Throughout malignant transformation, stromal ERα expression was progressively upregulated, while epithelial PR expression was downregulated at the transformed epithelium of the CIS and ICC stage—underscoring the carcinogenic role of stromal ERα and the tumor suppressor role of epithelial PRs. After complete transformation, stromal PRB levels were significantly increased, and its loss was associated with more significant metastatic potential and poorer prognosis. These findings provide a valuable foundation for developing targeted prevention and treatment strategies for cervical cancer.

## Figures and Tables

**Figure 1 ijms-26-04418-f001:**
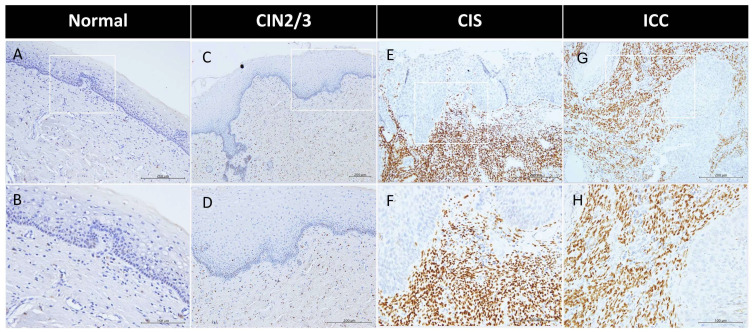
Progressive increase in stromal ERα in cervical cancer development, from normal to CIN2/3, CIS, and ICC cervical specimens. Representative IHC images of ERα in the normal cervix (**A**,**B**), CIN2/3 (**C**,**D**), CIS (**E**,**F**), and ICC (**G**,**H**) are shown. Magnification: 200× in (**A**,**C**,**E**,**G**); 400× in (**B**,**D**,**F**,**H**).

**Figure 2 ijms-26-04418-f002:**
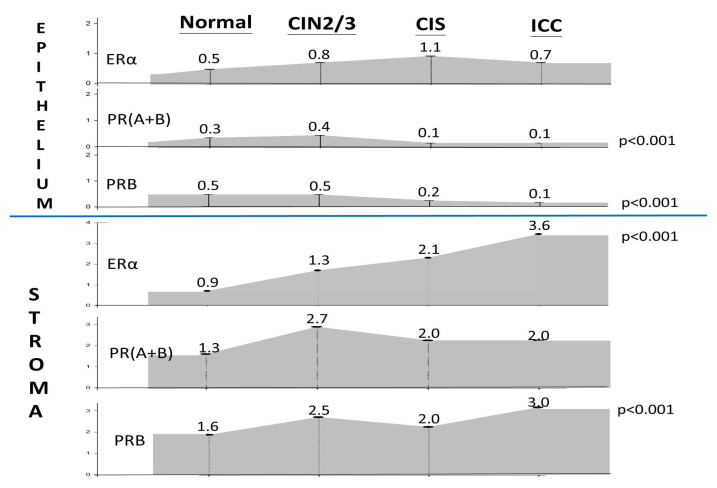
Changes in the average IRS of ERα and PR expression levels in the epithelium and stroma of normal, CIN2/3, CIS, and ICC cervical specimens.

**Figure 3 ijms-26-04418-f003:**
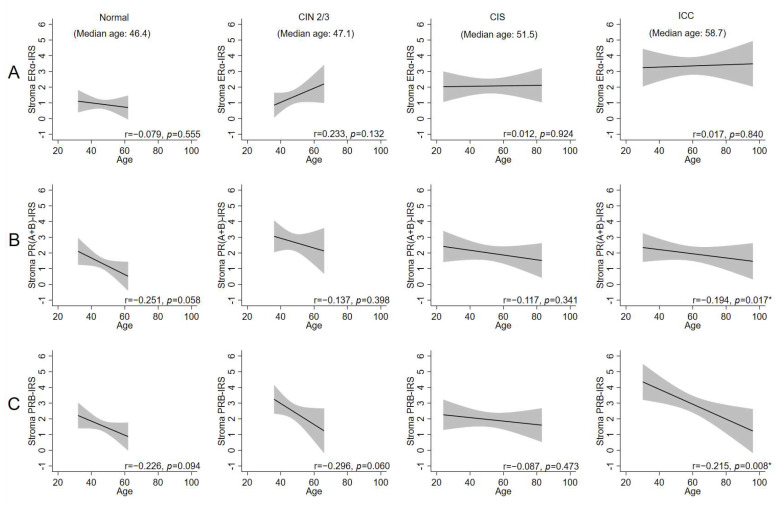
Age-specific IRS of (**A**) ERα, (**B**) PR(A+B), and (**C**) PRB expression levels in the stroma of normal, CIN2/3, CIS, and ICC cervical specimens. * indicates statistical significance at *p* < 0.05.

**Figure 4 ijms-26-04418-f004:**
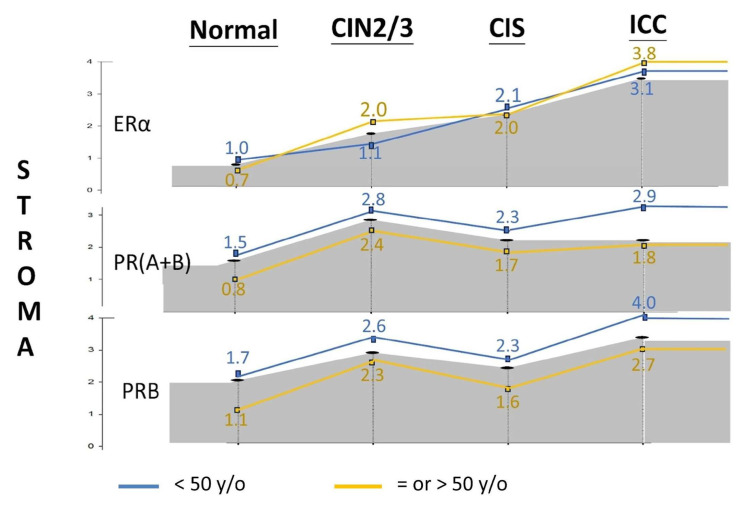
IRS of stromal PR expression levels in different severities of cervical neoplasia, stratified by age [blue line: <50 years old (y/o); yellow line: = or >50 y/o]. IRS, immunoreactive score.

**Figure 5 ijms-26-04418-f005:**
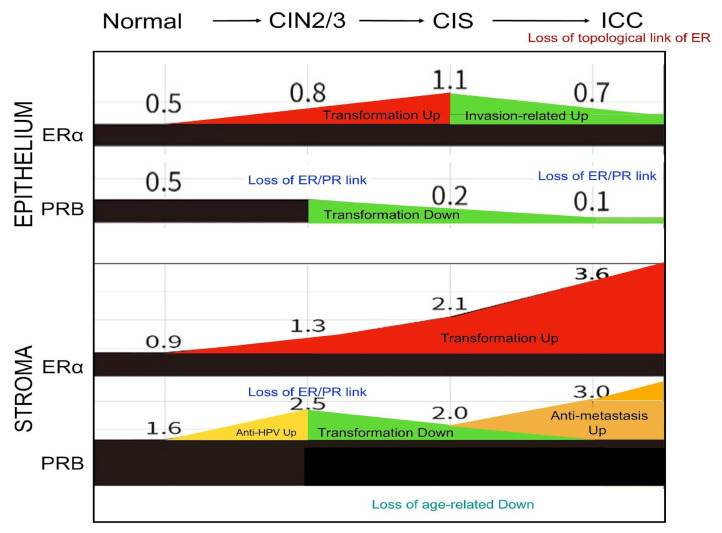
The possible significance of changes in ERα and PR expression levels in the epithelium and stroma of the cervix during malignant transformation. Increases (shown in red and yellow) and decreases (shown in green) in IRS at each stage of cervical carcinogenesis are presented, with potential significance noted. Moreover, the loss of coexpression links between ERα and PRs, the loss of topological coexpression of receptors in the epithelium and stroma, and age-related downregulation are highlighted.

**Figure 6 ijms-26-04418-f006:**
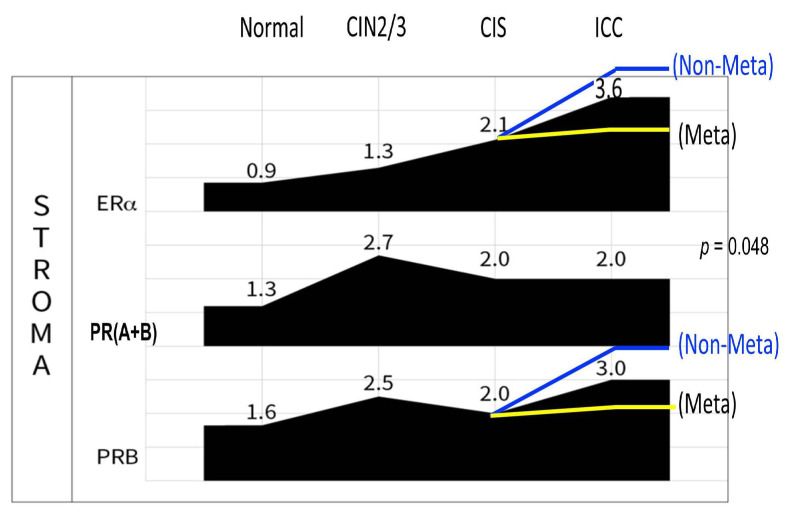
Difference of the stromal PRB IRS in distantly metastatic (yellow line) and non-metastatic (blue) ICCs. Risen ERα and PRB were associated with less distant metastasis.

**Figure 7 ijms-26-04418-f007:**
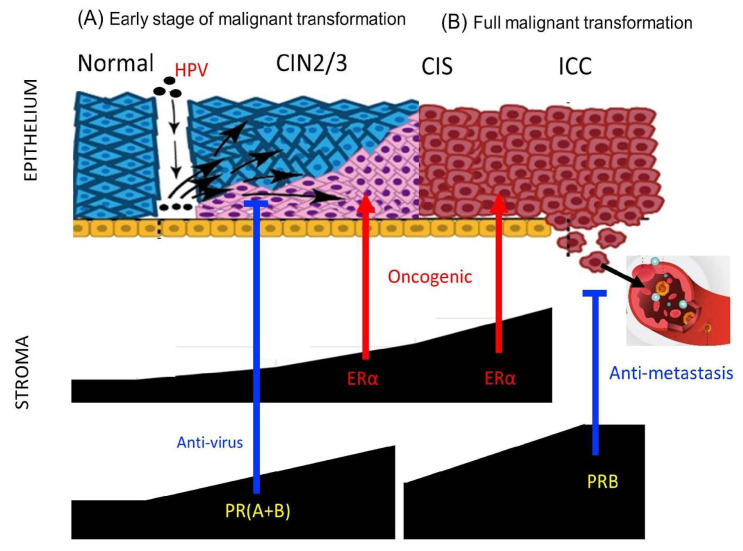
Possible roles of stromal P4/PR in HPV infection and cervical cancer progression. (**A**) At the early stage of malignant transformation (intraepithelial neoplasia), stromal progesterone and PRs promote antiviral innate immunity to counteract HPV infection, whereas stromal ERα is necessary for HPV-induced carcinogenesis. (**B**) After complete transformation or in invasive cancer, upregulated stromal PRB inhibits hematogenous metastasis. In cases where stromal PRs are not upregulated or are lost, distant metastasis may develop, which is associated with poor prognosis. Symbols: blue line: inhibition, red arrow: activation.

**Table 1 ijms-26-04418-t001:** Demographics of patients with Normal, CIN 2/3, CIS and ICC and the related ERα/PR expression (n = 331).

Characteristic	Normal	CIN2/3	CIS	ICC	*p* Value	*p* for Trend
Number	58	44	70	159		
Age (n = 330)	46.4 ± 6.4	47.1 ± 7.6	51.5 ± 15.1	58.7 ± 15.4	<0.001 *	
Age Group (n = 330)					<0.001 *	<0.001 **
<50 y/o	43 (74.1%)	30 (68.2%)	36 (51.4%)	56 (35.4%)		
≥50 y/o	15 (25.9%)	14 (31.8%)	34 (48.6%)	102 (64.6%)		
Parity Group (n = 256)					<0.001 *	<0.001 **
0	8 (14.0%)	4 (16.0%)	0 (0.0%)	5 (3.7%)		
1~2	29 (50.9%)	14 (56.0%)	8 (20.0%)	26 (19.4%)		
≥3	20 (35.1%)	7 (28.0%)	32 (80.0%)	103 (76.9%)		
Epithelium						
ERα expression, IRS (n = 313)	0.5 ± 0.7	0.8 ± 1.2	1.1 ± 1.4	0.8 ± 1.7	0.132	
positive rate (%)	17/55 (30.9%)	12/34 (35.3%)	43/70 (61.4%)	44/154 (28.6%)	<0.001 *	0.574 **
PR(A+B) expression, IRS (n = 307)	0.3 ± 0.6	0.4 ± 0.7	0.1 ± 0.4	0.1 ± 0.6	0.012 *	
positive rate (%)	11/54 (20.4%)	11/37 (29.7%)	6/68 (8.8%)	6/148 (4.1%)	<0.001 *	<0.001 **
PRB expression, IRS (n = 316)	0.5 ± 0.9	0.5 ± 0.8	0.2 ± 0.6	0.1 ± 0.4	<0.001 *	
positive rate (%)	14/53 (26.4%)	11/37 (29.7%)	11/70 (15.7%)	12/156 (7.7%)	<0.001 *	<0.001 **
Stroma						
ERα expression, IRS (n = 322)	0.9 ± 1.1	1.3 ± 1.5	2.1 ± 1.9	3.6 ± 3.3	<0.001 *	
positive rate (%)	30/58 (51.7%)	28/43 (65.1%)	57/70 (81.4%)	125/151 (82.8%)	<0.001 *	<0.001 **
PR(A+B)expression, IRS (n = 317)	1.3 ± 1.3	2.7 ± 1.7	2.0 ± 2.0	2.1 ± 2.3	0.009 *	
positive rate (%)	38/58 (65.5%)	36/40 (90.0%)	51/68 (75.0%)	98/151 (64.9%)	0.012 *	0.288 **
PRB expression, IRS (n = 320)	1.6 ± 1.3	2.5 ± 1.6	2.0 ± 1.9	3.2 ± 3.2	<0.001 *	
positive rate (%)	45/56 (80.4%)	38/41 (92.7%)	55/70 (78.6%)	118/153 (77.1%)	0.171 *	0.247 **

CIN: cervical intraepithelial neoplasia, CIS: Carcinoma in situ, ICC: Invasive cervical carcinoma, IRS: immunoreactive score = (Intensity × Percentage) of immunohistochemistry staining. Data are presented as number or mean ± standard deviation. * *p* for one-way ANOVA or Chi-squared test; ** *p* for the Cochran-Armitage trend test.

**Table 2 ijms-26-04418-t002:** Correlation of ERα vs. PR expression in the epithelium and stroma of normal cervix and different severity of cervical neoplasia.

Group	Stroma ERα vs. PR(A+B)				Stroma ERα vs. PRB				Epithelium ERα vs. PR(A+B)				Epithelium ERα vs. PRB		
	N	PCC	*p* Value		N	PCC	*p* Value		N	PCC	*p* Value		N	PCC	*p* Value
Normal	58	0.528	<0.001 *		56	0.589	<0.001 *		54	0.616	<0.001 *		53	0.555	<0.001 *
CIN2/3	39	0.292	0.072		41	0.253	0.111		34	0.534	0.001 *		33	0.335	0.057
CIS	68	0.607	<0.001 *		70	0.624	<0.001 *		68	0.288	0.017 *		70	0.389	0.001 *
ICC	142	0.467	<0.001 *		148	0.672	<0.001 *		145	0.047	0.572		151	0.101	0.219
Overall	307	0.458	<0.001 *		315	0.661	<0.001 *		301	0.177	0.002 *		307	0.206	<0.001 *

CIN2/3: cervical intraepithelial neoplasia grade 2 or grade 3, CIS: Carcinoma in situ, ICC: Invasive cervical carcinoma. PCC: Pearson’s correlation coefficient, IRS: immunoreactive score = (Intensity × Percentage) of immunohistochemistry staining. * *p* value < 0.05.

**Table 3 ijms-26-04418-t003:** Correlation of stroma vs. epithelium for ERα and PR in normal cervix and different severity of cervical neoplasia.

Group	Stroma vs. Epithelium ERα				Stroma vs. Epithelium PR(A+B)				Stroma vs. Epithelium PRB		
	N	PCC	*p* Value		N	PCC	*p* Value		N	PCC	*p* Value
Normal	55	0.326	0.015 *		54	0.478	<0.001 *		53	0.424	0.002 *
CIN2/3	34	0.373	0.030 *		35	0.046	0.794		37	0.285	0.087
CIS	70	0.316	0.008 *		68	-0.056	0.652		70	0.337	0.004 *
ICC	145	0.293	<0.001 *		146	0.056	0.499		147	0.024	0.773

CIN2/3: cervical intraepithelial neoplasia grade 2 or grade 3, CIS: Carcinoma in situ, ICC: Invasive cervical carcinoma. PCC: Pearson’s correlation coefficient, IRS: immunoreactive score = (Intensity × Percentage) of immunohistochemistry staining. * *p* value < 0.05.

## Data Availability

The original contributions presented in this study are included in the article. Further inquiries can be directed to the corresponding author.

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
