# Peer review of "Topological Distribution of the Sex Hormone Receptor Expressions Highlights the Importance of Stromal ERα and Epithelial PR in Malignant Transformation of the Uterine Cervix"

_ijms, 2025, doi:10.3390/ijms26094418_

Round 1

Reviewer 1 Report

Comments and Suggestions for Authors

The results from this study provide new insights into the hormonal roles in the development of cervical carcinomas through their precursor stages. The study design is robust, the manuscript is well-written, and the results are presented nicely. There are several issues that need to be fixed before publication.

  1. Fig. 1 c and d photomicrographs do not represent CIN2/3 lesions. They are normal squamous epithelium. Please consult a pathologist to replace them in this figure.
  2. CIS is an arcane term and is no longer used nowadays. It is equivalent to HSIL. Please also consider using the Bethesda grading system, HSIL and LSIL. CIN system is, again old terminology.
  3. The biological implications of the findings should be further articulated in the Discussion
  4. It is not sure if "ROC" needs to be shown in the authors' affiliations. It can be confused with another country and can be misleading.  

Author Response

Dear reviewer 1,

Thank you for your valuable comments, which make this study more readable and complete. We have made revisions accordingly. All revisions were highlighted in the manuscript; some deletions can only be seen in tracing mode.

We look forward to hearing from you at your earliest convenience.

Sincerely,

Tang-Yuan Chu, M.D., Ph.D.

Department of Obstetrics and Gynecology, Hualien Tzu Chi Hospital, Hualien

Centre for Prevention of Gynecological Cancer, Hualien Tzu Chi Hospital, Hualien, Taiwan, R.O.C

707, Sec. 3, Chung Yang Rd., Hualien City, Hualien 970, Taiwan, R.O.C.

Email: hidrchu@gmail.com ; Tel: +886-3-8561825#15610; Fax: +886-3-8577161

#Respond for Reviewer 1:

The results from this study provide new insights into the hormonal roles in the development of cervical carcinomas through their precursor stages. The study design is robust, the manuscript is well-written, and the results are presented nicely. There are several issues that need to be fixed before publication.

Comment: Fig. 1 c and d photomicrographs do not represent CIN2/3 lesions. They are normal squamous epithelium. Please consult a pathologist to replace them in this figure.

Response:=>We have consulted our pathologist and replaced Fig1c with new CIN2/3 pictures, as shown below, and revised figure1.

Comment:CIS is an arcane term and is no longer used nowadays. It is equivalent to HSIL. Please also consider using the Bethesda grading system, HSIL and LSIL. CIN system is, again old terminology.

Response:=>We are afraid that we cannot use the terms ‘LSIL’ or ‘HSIL’ in this study because the data of LSIL(CIN1) is unavailable. Actually, we purposefully avoid presenting LSIL because it indicates possible cervical dysplasia, a transient status more than likely caused by HPV.  Studies show that approximately 80% of LSIL cases regress within 2 years[1]. Because of this, LSIL/CIN1 results can be managed with a simple "watch and wait" philosophy. We use Normal as a base and CIN2/3, CIS and ICC to see the change from step to step.

References:

1.Maria Teresa Bruno , Nazario Cassaro , Francesca Bica , Sara Boemi. Progression of CIN1/LSIL HPV Persistent of the Cervix: Actual Progression or CIN3 Coexistence.

Infect Dis Obstet Gynecol. 2021 Mar 9;2021:6627531. doi: 10.1155/2021/6627531

Comment:The biological implications of the findings should be further articulated in the Discussion

Response:=> We added a paragraphy in the Discussion on page 11 to further articulate he biological implications of the findings:’ The findings of the present study suggest that cervical carcinogenesis is not purely an epithelial process, but involves significant stromal reprogramming and disruption of epithelial-stromal hormonal crosstalk. The alteration in ERα and PR expression patterns reflects a shift from hormone-regulated homeostasis to a dysregulated microenvironment that supports malignant progression. This has the following potential implications: First, therapeutic targeting of stromal ER signaling; Second, biomarker development for early detection or progression risk. Third, understanding age and hormone-related risk factors. Lastly, potential roles for hormonal manipulation (e.g., progestins) in prevention or treatment.’

Comment:It is not sure if "ROC" needs to be shown in the authors' affiliations. It can be confused with another country and can be misleading. 

Response:=> To avoid misleading we delete ‘ROC’ shown in the authors' affiliations.

Reviewer 2 Report

Comments and Suggestions for Authors

The authors described tissue-specific changes in expression levels of ERα and PR during cervical cancer development. It is clinically important and has high translational value. However, there are some concerns to be addressed. 

Results section 3.3: p values indicate the trend for PRA/B and PRB is significant only for ICC. The section should be revised to clearly convey this.

Figure 2: Several reports including one of references authors cited have shown that ERα expression is gradually decreased as the disease progresses to cancer. But the results presented here are different. It should be addressed/discussed.

Figure 4: it would be helpful to have epithelium data.

Figure 5: The epithelium ERα graph gives an impression that reduced ERα expression is associated with decreased invasion. As the authors stated, the loss of ERα expression enhances cervical cancer invasion. It should be revised.

The discussion section has paragraphs that are highly speculative and distantly related to the manuscript. For example, pg9, second paragraph describes antiviral effect of PR and MPA although the ms has no data related to HPV infection history of patients. Also, the last paragraph makes firm claims as if they are supported by functional data. These need be removed.

A few minor points are: 

-The instruction for Results section (pg 4) should be removed.

-Some references are incorrect (for example, 18, 19, 20, and 21). Correctness of all references need be confirmed.

-In Abstract, PRB IRS scores are not complete: only two values are described.

-ERα and ER are used interchangeably. For clarity, ERα should be used throughout the manuscript.

Comments on the Quality of English Language

There are some grammatical errors and some expressions are not clear. 

For example, pg2, second paragraph from the bottom: ".......treatment with MPA and 17β-estradiol prevented cervical cancer development......" gives impression that both MPA and estrogen inhibit cervical cancer. It should be revised. Also, "Topological transition" in the title could be misleading as there are no topological changes to the tissue or cells. 

It is recommended to have a professional editing service. 

Author Response

Dear reviewer 2,

Thank you for your valuable comments which make this study more readable and complete. We have made revisions accordingly. All revisions were highlighted in the manuscript; some deletions can only be seen in tracing mode.

We look forward to hearing from you at your earliest convenience.

Sincerely,

Tang-Yuan Chu, M.D., Ph.D.

Department of Obstetrics and Gynecology, Hualien Tzu Chi Hospital, Hualien

Centre for Prevention of Gynecological Cancer, Hualien Tzu Chi Hospital, Hualien, Taiwan, R.O.C

707, Sec. 3, Chung Yang Rd., Hualien City, Hualien 970, Taiwan, R.O.C.

Email: hidrchu@gmail.com ;Tel: +886-3-8561825#15610 ; Fax: +886-3-8577161

#Respond for Reviewer 2:

The authors described tissue-specific changes in expression levels of ERα and PR during cervical cancer development. It is clinically important and has high translational value. However, there are some concerns to be addressed.

Comment: Results section 3.3: p values indicate the trend for PRA/B and PRB is significant only for ICC. The section should be revised to clearly convey this.

Response:=> Thank you for your heedful review. We revised this section on page 6: ‘We determined whether the expression of hormone receptors changes with age(Figure 3A–3C). In the stroma, there was a statistically non-significant an observable trend of age-related reduction in PR(A+B) and PRB levels in normal and CIN2/3 specimens. This trend was lost in CIS but became statistically significant and reappeared in ICC (Figure 3A–3C)….’

Comment: Figure 2: Several reports including one of references authors cited have shown that ERα expression is gradually decreased as the disease progresses to cancer. But the results presented here are different. It should be addressed/discussed.

Response:=> In our literature search, we found only one immunohistochemistry (IHC) study examining ERα expression across different severities of cervical neoplasia[1].  This 1989 study included only 44 cervical tissue samples and found that epithelial ERα expression is gradually decreased as the disease progresses to cancer without specifically examining carcinoma in situ (CIS). While in this  study, epithelial ERα expression is gradually increased from normal, CIN2/3 to CIS and decreased at CIS-ICC progression. The difference may be explained by different sample sizes, methodologies and patients' menstrual cycle statuses, as shown in the table below.The sample size of the previous study was limited, and only LSIL, HSIL and ICC cases were assessed. In this study, as we mentioned before we purposefully avoid presenting LSIL because it indicates possible cervical dysplasia, a transient status; and we specifically examine carcinoma in situ (CIS), where moderate ER expression was observed in the present study. We utilize clinical-grade antibodies for IHC staining to ensure accuracy and reliability.

PMC2913367

Our study

Sample size & grouping

Normal =22;LSIL= 12;HSIL= 13;SCC =27

Normal=58;CIN1/LSIL=0(avoided);CIN2/3= 44;CIS= 70;SCC= 159

Immunohistochemical Analysis

Sections were incubated with mouse monoclonal anti-EZH2 (1:250 dilution, BD Biosciences, San Jose, CA) or mouse monoclonal anti-ESR1 (1:50 dilution, Dako, Carpinteria, CA) overnight at 4°C. Slides were washed in PBS, then incubated with a biotinylated horse anti-mouse secondary antibody for 30 minutes at room temperature. Antigen-antibody complexes were detected with the avidin-biotin peroxidase method using Vector diaminobenzidine as a chromogenic substrate (Vectastain ABC kit, Vector Laboratories).Immunostained sections were lightly counterstained with hematoxylin

The slides were incubated with a peroxidase block to neutralize endogenous peroxidase activity. Subsequently, the slides were treated with either anti-ERα monoclonal antibody (dilution 1:250; ab108398; Abcam) or anti-PR antibody (PRB-specific YR85; dilution 1:100; ab32085; Abcam; PRA-specific NCL-L-PGR-312, which was found also binds PRB, hence marked as PR(A+B)[14, 15][6, 11]; dilution 1:100; PGR-312-L-F; Leica Biosystems) for 30 min[14, 16] [6, 12]. Next, the slides were incubated with the Novolink polymer and then treated with DAB chromogen solution to visualize peroxidase activity.

Staining evaluation &

Defination of positivity

Four tiered scale for intensity (−, absent; ±, weak; +, moderate; ++, strong) in the cell nucleus. (−) or (+/−) staining was considered negative, while (+) or (++) was considered positive.

The results were evaluated using the immunoreactive score (IRS).The percentage of positively stained cells was scored as: 0 (0%), 1 (1%–10%), 2 (11%–50%), 3 (51%–80%), or 4 (>80%). The staining intensity was scored as: 0 (none), 1 (weak), 2 (moderate), or 3 (strong). Multiplying these two scores yielded an IRS of 0 to 12. IRS >0.5 was considered positive.

On the other hand, the epithelial ERα expression of the squamous epithelium depends upon the menstrual cycle: in the early proliferative phase cells of all layers are negative. In the midphase of proliferation the basal and parabasal layers become positive, and in the secretory phase positive cell nuclei can be found up to the superficial layers[2]. In this study, most patients in precancerous groups were in premenopausal status, i.e. 74.1%,68.2% and 51.4% in Normal, CIN2/3 and CIS groups, respectively. Their menstrual cycle status at conization were not available, which may be considered a reasonable selection bias. We made a summary and added it to the Discussion on page 9.

Reference:

  1. Yali Zhai 1, Guido T Bommer, Ying Feng, Alexandra B Wiese, Eric R Fearon, Kathleen R Cho

. Loss of estrogen receptor 1 enhances cervical cancer invasion. Am J Pathol 2010 Aug;177(2):884-95. doi: 10.2353/ajpath.2010.091166. Epub 2010 Jun 25.

  1. D S Mosny 1, J Herholz, W Degen, H G Bender. Immunohistochemical investigations of steroid receptors in normal and neoplastic squamous epithelium of the uterine cervix

Gynecol Oncol.1989 Dec;35(3):373-7. doi: 10.1016/0090-8258(89)90082-6.

Comment: Figure 4: it would be helpful to have epithelium data.

Response: =>We added this data as supplementary  Table S2

Comment: Figure 5: The epithelium ERα graph gives an impression that reduced ERα expression is associated with decreased invasion. As the authors stated, the loss of ERα expression enhances cervical cancer invasion. It should be revised.

Response:=> Yes, the loss of ERα expression enhances cervical cancer invasion. Figure 5 was revised.

Comment: The discussion section has paragraphs that are highly speculative and distantly related to the manuscript. For example, pg9, second paragraph describes antiviral effect of PR and MPA although the ms has no data related to HPV infection history of patients. Also, the last paragraph makes firm claims as if they are supported by functional data. These need be removed.

Response:=> Although we don’t have further convicing data, we really wish to discuss or share these thinkings to the reader or other researchers for induction or enhancement of  further investigation in this field. We removed the words or sentences that are highly speculative and distantly related to the manuscript—‘This observation supports the notion that progesterone plays an antiviral and anti-transformation role.’

We revised some descriptions of these parts to be more softened and more objective. For example: ‘Thus, progesterone may exert an antiviral effect through its action on PRs in the stroma during the early stages of cervical transformation with persistent HPV infection (Figure 5)’ was changed to ' Whether progesterone may exert an antiviral effect through its action on PRs in the stroma during the early stages of cervical transformation with persistent HPV infection (Figure 5) is worth further investigated.’  

‘Our IHC study of the human cervix corroborates this finding,…’was changed to ‘Our IHC study of the human cervix echoes this finding,…’

‘In the normal to CIN2/3 stage, stromal PR increases to combat HPV infection and transformation.’  was changed to ‘It is speculated that the normal to CIN2/3 stage, stromal PR increases to combat HPV infection and transformation.’

‘Moreover, we discovered that although cervical carcinomas maintained consistent levels of ERα across different ages, stromal PRB expression  exhibit      a steady decline with advancing age at diagnosis.’  was changed to‘Moreover, we discovered that although cervical carcinomas maintained consistent levels of ERα across different ages, stromal PRB expression seemed to exhibits a steady decline with advancing age at diagnosis.’

We also emphasized that ‘However, further studies are warranted to prove this speculation’ on page 10.

The last paragraph 5. The conclusion part was removed. It was rewritten as follows: ‘This study offers new insights into the roles of sex hormone receptors and their interactions between the epithelium and stroma during the development of cervical carcinomas from precursor stages. Throughout malignant transformation, stromal ERα expression was progressively upregulated, while epithelial PR expression was downregulated at the CIS-ICC stage—underscoring the carcinogenic role of stromal ERα and the tumour suppressor role of epithelial PRs. After complete transformation, stromal PRB levels were significantly increased, and its loss was associated with greater metastatic potential and poorer prognosis. These findings provide a valuable foundation for developing targeted prevention and treatment strategies for cervical cancer.’on page 11.

A few minor points are:

Comment: The instruction for Results section (pg 4) should be removed.

Response:=>It was removed.

Comment: Some references are incorrect (for example, 18, 19, 20, and 21). Correctness of all references need be confirmed.

Response:=> The references 18,19,20,21 are checked, 18,19 were errors due to transferring text to the template of IJMS and they were corrected. The other references were also carefully checked, a few more errors were found, and all the errors were also corrected. Few less relevant references were changed to more relevant references.

Comment: In Abstract, PRB IRS scores are not complete: only two values are described.

Response:=>It was revised.

Comment:ERα and ER are used interchangeably. For clarity, ERα should be used throughout the manuscript.

Response:=>ERα was used throughout the revised manuscript.

Comments on the Quality of English Language

There are some grammatical errors and some expressions are not clear.

Comment:For example, pg2, second paragraph from the bottom: ".......treatment with MPA and 17β-estradiol prevented cervical cancer development......" gives impression that both MPA and estrogen inhibit cervical cancer. It should be revised.

Response:=>The expression of progesterone receptor is estrogen-dependent[1], the use of 17β-estradiol in the study was actually to induce progesterone receptor and follow by progesterone treatment and its effect. To avoid confusion, we revised this sentence to ‘In transgenic mice with cervical intraepithelial neoplasia (CIN) lesions, treatment with medroxyprogesterone acetate (MPA) was showed to prevent development of cervical cancer.’ on page 2.

Reference:

  1. Flototto, T., et al., Molecular mechanism of estrogen receptor (ER)alpha-specific, estradiol-dependent expression of the progesterone receptor (PR) B-isoform. J Steroid Biochem Mol Biol, 2004. 88(2): p. 131-42.

Comment: Also, "Topological transition" in the title could be misleading as there are no topological changes to the tissue or cells. 

Response:=>To avoid misunderstanding, the title was revised to ‘Topological distribution of the sex hormone receptor expressions highlights the importance of stromal ERα and epithelial PR in malignant transformation of the uterine cervix’.

Comment: It is recommended to have a professional editing service.

Response:=>The manuscript was revised to correct spelling errors, to improve grammar and fluency of the manuscript.
